# Co-Encapsulation of Curcumin and Diosmetin in Nanoparticles Formed by Plant-Food-Protein Interaction Using a pH-Driven Method

**DOI:** 10.3390/foods12152861

**Published:** 2023-07-27

**Authors:** Chong Yu, Jingyu Shan, Ze Fu, Hao Ju, Xiao Chen, Guangsen Xu, Yang Liu, Huijing Li, Yanchao Wu

**Affiliations:** 1Harbin Jilida Technology Co., Ltd., Harbin 150001, China; yc15776578163@163.com; 2School of Marine Science and Technology, Harbin Institute of Technology, Weihai 264209, China

**Keywords:** zein, soy protein isolate, curcumin, diosmetin, co-assembly, nanoparticles, pH-driven method

## Abstract

In this work, a pH-driven method was used to prepare zein–soy protein isolate (SPI) composite nanoparticles (NPs). The mass ratio of SPI to zein influenced the Z-average size (Z-ave). Once the zeta potential stabilized, SPI was completely coated on the periphery of the zein NPs. The optimal mass ratio of zein:SPI was found to be 2:3. After determining the structure using TEM, curcumin (Cur) and/or diosmetin (Dio) were loaded into zein–SPI NPs for co-encapsulation or individual delivery. The co-encapsulation of Cur and Dio altered their protein conformations, and both Cur and Dio transformed from a crystalline structure to an amorphous form. The protein conformation change increased the number of binding sites between Dio and zein NPs. As a result, the encapsulation efficiency (EE%) of Dio improved from 43.07% to 73.41%, and thereby increased the loading efficiency (LE%) of zein-SPI NPs to 16.54%. Compared to Dio-loaded zein–SPI NPs, Cur/Dio-loaded zein–SPI NPs improved the storage stability of Dio from 61.96% to 82.41% within four weeks. The extended release of bioactive substances in the intestine during simulated gastrointestinal digestion improved the bioavailability. When exposed to a concentration of 0–800 µg/mL blank-loaded zein–SPI NPs, the viability of HepG2 and LO-2 cells was more than 90%, as shown in MTT assay tests. The zein–SPI NPs are non-toxic, biocompatible, and have potential applications in the food industry.

## 1. Introduction

As a common food additive, curcumin (Cur) has been widely used because of its efficacy. Cur is mainly derived from turmeric (*Curcuma longa* L.), and turmeric was traditionally used in ayurvedic medicine [1]. Cur as a hydrophobic flavonoid is an effective bioactive compound and has a variety of bioactive properties such as anti-inflammatory, anti-oxidant, anti-tumor, anti-bacterial and other effects [2,3,4]. Diosmetin (Dio) is a flavonoid glycoside isolated from citrus fruits and could be used to treat vascular diseases. The bioactive properties of Dio include anti-inflammatory and antioxidant effects as well as alleviating diabetic symptoms and hyperlipidemia, while the toxicological studies have shown that it has good biosafety in a limited concentration range [5]. Dio can reduce damage to the intestinal epithelial barrier and the damage caused by oxidative stress, and it regulates the intestinal microbial flora and thereby helps to fight colitis by reducing the degree of oxidative stress and re-regulating the microenvironment in the intestine [6]. However, due to the existence of the first pass effect of bioactive substances and drastic pH changes in the stomach, a premature release in the stomach results in a large loss. How to ensure these health-promoting substances are taken up in the intestine, that is, how to ensure the effect of a slow release of these bioactive substances, has become a problem that we need to solve [7]. At the same time, the hydrophobicity, poor chemical stability, photoinstability and poor bioavailability of polyphenols also limit their further application, and thus a variety of delivery systems have been developed, including liposomes, emulsions, gels and electrosprays [8,9,10]. It is known that co-delivery of different bioactive substances might enhance the effect of the application of both. For example, the cell viability of MCF-7 breast cancer cells exposed to Cur and quercetin co-encapsulated in casein micelles at a concentration of 0–900 µg was lower than that of both bioactive substances encapsulated alone and of cancer cells treated with free Cur and quercetin. Therefore, the co-encapsulation of Cur and quercetin increases their potential ability to inhibit MCF-7 breast cancer cells growth [11]. In Hong’s work [12], the anti-inflammatory activity is enhanced by a synergistic effect of Cur and apigenin. As proven by Chen [13], the storage stability and the slow release effect of Cur were improved by the co-encapsulation of Cur and piperine. Although there are many co-encapsulated reports, the co-encapsulated of Cur and Dio is still unexplored and thus became the research interest of this work.

zein is an excellent food raw material extracted from corn starch, rich in protein, and is recognized as an advantageous food material by the Food and Drug Administration of the United States. In recent years, it has been widely used in food nanoparticles (NPs), food films, fibers, and so on. When ethanol is used as a solvent in certain proportions, it has the advantages of being applicable in edible film synthesis [14,15]. The structure of zein is a hairclip with an extremely high content of non-polar amino acids. It is a hydrophilic intermediate structure with hydrophobic ends [16]. Therefore, the formed NPs exhibit strong hydrophobicity, and thus can be used as new food-based nanocarriers [17]. To deliver hydrophobic bioactive substances by nanocarriers, common preparation methods include anti-solvent precipitation, liquid–liquid dispersion and so on [18]. However, the presence of high temperatures, salts, acids and alkali or different isoelectric points can affect the stability of zein NPs, causing them to aggregate and precipitate [19]. 

In order to stabilize zein NPs, stabilizers such as proteins and polysaccharides have been applied to the surface of zein NPs in recent studies. For example, zein and whey protein complex NPs have been constructed by Liu, where the whey protein interacted with zein through hydrogen bonding and increased the stability of zein NPs [20]. Chen used an anti-solvent method to wrap lactoferrin around zein to construct a binary colloidal complex for the delivery 7,8-dihydroxyflavone, whereby the addition of lactoferrin improved the physical properties of zein, improved the drug encapsulation efficiency, and increased its bioavailability [21]. Some polysaccharide components extracted from natural plants, such as fucoidan, can be used as a stabilizer, as was researched by Liu et al. who co-assembled fucoidan with zein for covering the surface of NPs. Due to co-assembly, zein’s isoelectric point (PI) was changed and the stability of NPs at neutral pH increased. At the same time, the presence of hydrophilic polysaccharide fucoidan captured the bioactive substance on a more hydrophilic shell and increased the overall encapsulation efficiency (EE%) of NPs, and the use of fucoidan increased the resistance of zein to the external environment, such as improving its ability to withstand ions, resulting in fewer problems of poor stability and low loading of bioactive substances [22]. The phenomenon could be ascribed to electrostatic repulsion, steric stabilization, or a combination of both caused by relatively hydrophilic proteins and polysaccharide on the zein NPs surfaces [23,24]. Thus, the addition of stabilizers might result in zein complexes showing better physicochemical properties than zein NPs alone.

Traditionally, the methods for preparing zein NPs, such as the anti-solvent method and liquid-liquid dispersion method, all use organic solvents in practical operation. Accordingly, zein is dissolved in organic solvents because of its robust water insoluble property, and then the organic solvent in the mixed solution is removed by rotating evaporation after mixing with water to form zein NPs [21]. Common organic solvents used to prepare zein NPs include methanol, ethanol and isopropanol, in which the most commonly used solvents are 60–80% ethanol solutions [25]. However, excessive intake of ethanol is harmful to human health. In addition, ethanol is a flammable and explosive substance, which brings challenges to its transportation. Moreover, due to the high cost of ethanol in actual production, the zein NPs prepared by this method are not suitable for commercialization and industrialization. In recent years, with the deepening of people’s concept of health, people’s demand for non-alcoholic food is increasing, so the application of this traditional organic solvent to prepare zein NPs has become increasingly narrow [26].

With the continuous in-depth research on the preparation methods of zein NPs, researchers have developed a new method for preparing zein NPs, namely the pH-driven method [27]. By dissolving zein in a high concentration of alkaline solution and adjusting the pH values, the solubility of zein molecules decreases to form a hydrophobic structure. While forming spherical structure, some bioactive substances that can be dissolved in alkaline but do not lose activity in a short time can be encapsulated in hydrophobic nanostructures. Cur and Dio can be deprotonated in alkaline solution because they contain hydroxyl groups and can be easily dissolved in sodium hydroxide solution and co-dissolved with zein, and then embedded into zein NPs under pH changes [28]. Compared with the traditional anti-solvent method and liquid–liquid dispersion method, the pH-driven method has the advantages of avoiding the use of organic solvents and simple preparation method, and is conducive to the subsequent industrialization and commercialization. Some subsequent researchers found further advantages of pH-driven method in their research, such as smaller particle size and better dispersion of NPs prepared by the pH-driven method, compared with the traditional anti-solvent method. As described in Chen’s article, foxtail millet prolamin (FP) and casein NPs prepared by the pH driven method have a smaller particle size compared with FP casein NPs prepared by the anti-solvent method [23].

As a plant food protein, soy protein isolate (SPI) has many active functional groups, such as hydrogen bonds, which can be combined through covalent or non-covalent bonds to form nanocarriers, for example, composite NPs formed by cross-slinking SPI and fucoidan [29]. Protein–protein interactions are common, such as zein and casein or bovine serum albumin (BSA), for the construction of NPs by the pH-driven method [30]. However, less has been reported on zein interactions with plant food proteins. Herein, using SPI as a stabilizer to produce zein–SPI composite NPs was reported for the first time. As a commonly used method for the fabrication of zein-based NPs, the pH-driven method has been used for the entrapment of curcumin, rutin, thymol astaxanthin and pterostilbene [31]. However, the Dio encapsulation by the pH-driven method has not been studied. The co-encapsulation of two different bioactive substances with the use of the anti-solvent method has been widely studied, and the exiting reports indicated that co-encapsulation could improve storage stability in comparison with single encapsulation. Chen and coworkers [26] observed that co-encapsulation of resveratrol with Cur resulted in extended stability of resveratrol from 30% to over 90%. We hypothesized that SPI could be co-assembled with zein by the pH-driven method to form zein-SPI NPs, and then Cur and Dio would be loaded into zein-SPI NPs by co-encapsulation technology. By the co-encapsulation of two different bioactive substances, the storage stability and release efficiency may be different from those loaded with individual bioactive substance alone.

For this reason, this work investigated the co-encapsulation of Cur and Dio with the use of zein and SPI as the materials. The pH-driven method was used to prepare the particles, and the specific structure of NPs formed was observed by TEM. The loading of two bioactive substances was carried out after particle size analysis and micromorphological characterization to determine the optimal NPs morphology. Subsequently, the NPs in the co-encapsulation of bioactive substances was explored by infrared, fluorescence spectroscopy, XRD and in vitro digestion release. Finally, the biocompatibility of blank NPs was evaluated using the MTT method. This work will provide a new method for using food materials to deliver bioactive substances, and the prepared NPs are non-toxic and biocompatible, and thus have potential applications in the food industry.

## 2. Material and Methods

### 2.1. Materials

zein and soy protein isolate were purchased from Shanghai Macklin Biochemical in China. Cur (purity ≥ 95%) and Dio (purity ≥ 98%) were purchased from Energy Chemical Company in Shanghai (China). Dulbecco-modified Eagle medium (DMEM), fetal bovine serum (FBS) and 3-(4,5-dimethyl-2-thiazolyl)-2,5-diphenyl-2-Htetrazolium bromide (MTT) were purchased from Solarbio Life Sciences Company (Beijing, China). The LogP values in this work can be found in Drugbank database (https://go.drugbank.com/) (accessed on 29 June 2023). Other reagents and solvents are standard, all in analytical grade.

### 2.2. Samples Preparation

Firstly, we prepared the zein stock solution (10 mg/mL) with different quantities of SPI (0.5, 1, 2, 3 and 4 mg/mL) in 25 mmol/L NaOH solution. The solution required magnetic stirring until the complete dissolution of zein and SPI. For the preparation of blank-loaded, zein-SPI NPs, the pH-driven method was followed by previous research with little adjustments [27]. Suffice to say, 2 mL of zein was added into 10 mL SPI solutions of different quantities. The zein–SPI mixture was stirred for 5 min, after which its pH values were adjusted from 12 to 7 with 0.5 mol/ L citric acid to obtain the blank-loaded, zein-SPI NPs. The preparation procedure of bioactive substance-loaded, zein-SPI NPs was the same as mentioned previously. Before the addition of zein in SPI, 3 mg/mL Cur or/ and Dio were dissolved in zein stock solution by stirring for 30 min until complete dissolution (the ratio of Cur: Dio was 1:1). The fabrication processes of Cur-, Dio-, and Cur/Dio-loaded, zein-SPI NPs were like that of blank zein-SPI NPs dispersion. All dispersions were adjusted to pH values from 12 to 7, and then centrifuged at 5000 rpm for 10 min to dislodge unencapsulated bioactive substances and large particles. We then removed some of the NPs from each dispersion group and lyophilized them for subsequent characterization analysis. The lyophilization process was performed using a freeze dryer (BoyiKang FD-1A-80, Beijing, China) and the temperature under the drying process was maintained at −79 °C.

### 2.3. Different Characterizations of zein-SPI NPs

#### 2.3.1. Particle Characterization

Characterization of bioactive substance-loaded, zein-SPI NPs was performed using a Zetasizer Nano-ZS90 (Malvern, Worcestershire, UK). Before the analysis, 1 mL of the sample was taken and mixed with ultrapure water at a 1:10 (*v*/*v*) ratio to avoid inaccurate results due to scattering effects. All data were measured three times and averaged. After the instrument was turned on, it was preheated for 10 min to prevent deviation in the results caused by environmental factors. The scattering angle was 90° and the operating temperature was around 25 °C. The repeatability error system was less than or equal to 1%. Finally, the three independent tests were conducted for mean ± SD expression.

#### 2.3.2. TEM

The diluted nanodispersions were prepared using a transmission electron microscope (JEOL-2100 microscope) at 200 kV. A total of 10 uL of the samples was dropped onto the plasma (glow power generation) carbon film network, after the samples were naturally dried using 2% (*w*/*v*) phosphotungstic acid as the stain and negatively stain different nano-dispersion for 1 min [23]. 

#### 2.3.3. FTIR

The FTIR spectra of different complex NPs were tested with the Nicolet 380 attenuated total reflectance (Thermo Electron, Waltham, MA, USA), collecting 32 scans ranging from 800 to 4000 cm^−1^ with a resolution of 4 cm^−1^. The spectral data were analyzed by Origin 2021 software.

#### 2.3.4. Fluorescence of Free Bioactive Substance and Encapsulate Bioactive Substance

The concentration of free Cur and Dio were 2 mg dissolved in 10 mL of 70% aqueous ethanol, and Cur as a single group was 2 mg dissolved in the same volume of 70% ethanol solution. The Cur- and Cur/Dio-loaded zein-SPI NPs were selected for fluorescence measurements on a Hitachi F-7000 Fluorescence Spectrophotometer (Tokyo, Japan). The emission spectra of Cur were obtained from 450 to 750 nm and excitation was set at 425 nm.

#### 2.3.5. XRD

After the nano-dispersion being centrifuged, the supernatant was lyophilized. The different samples included two single ingredients (free Cur and free Dio), and four nano-dispersion NPs (blank-, Cur-, Dio-, and Cur/Dio-loaded zein-SPI NPs) were determined using a DanDongHaoYuan DX2700 X-ray diffractometer (Liaoning, China). The data were collected in the range of 10° to 50° (2^θ^) in steps of 0.02°, respectively.

### 2.4. pH Stability

The pH stability of bioactive substance-loaded zein-SPI NPs composite was used according to a previously published work with some adjustments [2]. Concretely, the new bare Cur/Dio-loaded zein-SPI NPs were diluted 10-fold using HCl and NaOH (4 mol/L) to adjust the pH values from 3 to 9. After 6 h of storage, both particle diameter and Zeta-potential were used to test the nano-dispersion data.

### 2.5. Encapsulate and Loading

The bioactive substances were encapsulated after centrifuging at 5000 rpm for 10 min, and the supernatant of nano-dispersion was tested 15 times by a UV-gradiometer through dilution with dimethyl sulfoxide (DMSO). After mixing, the absorbance was measured at 426 nm and 348 nm for Cur and Dio, respectively. The concentrations of Cur were estimated based on standard curve in the range of 1.5–6 µg/mL (Y = 0.1643X R^2^ = 0.9999), and the concentration of Dio was in the range of 1.7–8.8 µg/mL (Y = 0.0618X R^2^ = 0.9998). To prevent mistakes, a HPLC Shimada 20AT C18 column (4.6 × 250 mm, 5 um) was used with methanol and distilled water (55:45) as the mobile phase, and we maintained the sample injection at 10 µL every time under the isocratic condition at 0.7 mL/min, in order to separate the bioactive substance.
(1)$$EE\left(\%\right)=\frac{\left(Total\;Bioactive-Free\;Bioactive\right)}{Total\;Bioactive}\times 100\%$$
(2)$$LE\left(\%\right)=\frac{\left(Total\;Bioactive-Free\;Bioactive\right)}{Total\;Weight\;of\;zein\;SPI\;NPs}\times 100\%$$

### 2.6. Storage Stability

The bioactive substance in nano-dispersion retention rate was evaluated to obtain the reality nano-dispersion potential storage time [24]. The fresh samples, including Cur-, Dio-, and Cur/Dio-loaded zein-SPI NPs, were put in a 4 °C environment for one month. The initial sample’s EE% and loading efficiency (LE%) values were tested before storage, after which all the samples’ EE% and LE% values were tested each week using the method mentioned in Section 2.4.

### 2.7. Bioactive Substance Release

#### 2.7.1. In Vitro Gastrointestinal Digestion

The in vitro gastrointestinal digestion operation was performed using the previous procedure with some modifications [32]. We first prepared simulated gastric fluid (SGF) and simulated intestinal fluid (SIF). SGF, 3.2 mg/mL pepsin and 2 mg/mL NaCl were adjusted to a pH of 2 using HCl solutions. SIF, 12 mg/mL bile salt, 2 mg/mL pancreatin, and 8.8 mg/mL NaCl were mixed and adjusted to a pH of 7.5 using NaOH solutions. After preparing SGF and SIF, 2 mL of each nanoparticle was added to 20 mL of SGF and digested at 37 °C for 90 min, followed by the addition to SIF and conducting further digestion for 270 min [30]. Every 30 min, analysis of the Z-average was performed at different digestion stages. At the same time, the bioactive substances released from the supernatant were collected to evaluate the different nano-dispersion bioactive substances’ situation.

#### 2.7.2. MTT

To evaluate the biosecurity of blank-loaded, zein-SPI NPs, it is necessary to examine their cytotoxicity. In this study, the MTT assay was carried out using L-O2 and HepG2 cells. Briefly, L-O2 and HepG2 cells were seeded in 96-well plates and incubated until the cell number reached around 70%. Various concentrations of the NPs (100, 200, 400 and 800 μg/mL) were added and incubated for 24 h at 37 °C. MTT (5 mg/mL) was then added and incubated for an additional 4 h (control group was also included). Finally, DMSO was added to stop the incubation, and the absorbance of each group was measured using a microplate reader at 490 nm to determine cell viability.

### 2.8. Re-Dispersibility of zein-SPI NPs

The lyophilized samples, including Cur-, Dio- and Cur/Dio-loaded zein–SPI NPs, were re-dispersed in ultrapure water. The concentrations of the re-dispersed solutions were the same as the freshly prepared nano-dispersion. The bioactive substance content, EE% and LE%, Z-average and zeta potential of the re-dispersed NPs were then measured.

### 2.9. Statistical Analysis

All data were measured at least three times. Data are presented as mean ± standard deviation (SD). Significant difference was calculated using the One-way ANOVA test through Prism 7.0 software. The *p* value was less than 0.05.

## 3. Results and Discussion

### 3.1. Co-Assembled zein-SPI Complex NPs

In previous experiments, appropriately proportioned shell thickness could construct relatively compact NPs. To our advantage, after achieving the NPs morphology, the NPs exhibited superior physicochemical properties [33]. In this experiment, to construct relatively stable, blank-loaded, zein-SPI NPs at a suitable ratio, SPI was used as a stabilizer wrapped around the periphery of hydrophobic zein NPs. The Z-average and zeta potential of the bare zein NPs were 153.3 nm and −48 mV, respectively, which are the same as previously reported results [2]. As shown in Figure 1, when the concentration of zein to SPI was 4:1 and 2:1, the Z-average of NPs continued to increase. Due to the low concentration of SPI, the NPs lacked sufficient electrostatic- and spatial-repulsive forces, so SPI only accumulated on the hydrophobic-zein molecules. With the increase of the concentration to 1:1, both the Z-average and zeta potential changed drastically, with the Z-average decreasing from 183.60 nm to 96.93 nm. The size of the constructed zein-SPI NPs was smaller than that of zein NPs, indicating that SPI may exist as a new stabilizer for zein NPs. When the concentration ratio was raised to 2:3, the nanoparticle size decreased from 96.93 nm to 81.73 nm, while its zeta potential remained almost unchanged at around −39.3 mV, which is consistent with the potential of SPI in a neutral aqueous solution (−39.3 mV). As the concentration ratio continued to increase, the excess SPI, which wrapped itself in the outer layer of the zein-SPI NPs, caused the nanoparticle size to also increase. Therefore, the optimal mass ratio (2:3) of zein–SPI was chosen as the final ratio for subsequent experiments, as zein-SPI NPs had the smallest Z-average and the most stable Z-average at this ratio.

### 3.2. TEM

From Figure 2A–F, it can be observed that both the blank-loaded zein-SPI NPs and Cur/Dio-loaded zein-SPI NPs have a Z-average in the range of 60–160 nm, which is consistent with what we observed using a particle size analyzer. It can be observed that the NPs were evenly distributed and spherical. 

From Figure 2C,F, it can be clearly observed that the shell thickness of the blank-loaded, zein-SPI NPs is significantly smaller than that of Cur/Dio-loaded zein-SPI NPs. In subsequent tests, it was found that the Z-average of Cur/Dio-loaded zein-SPI NPs was slightly higher than that of blank-loaded zein-SPI NPs, which is consistent with the result obtained by TEM. After loading Cur/Dio, the wall layer of SPI became irregular (see the arrow in the figure), and it was also found that the core color of the Cur/Dio-loaded zein-SPI NPs was darker and thicker than that of the blank-loaded zein-SPI NPs, indicating that Cur/Dio was successfully loaded into zein-SPI NPs.

### 3.3. FTIR

From Figure 3A, it can be observed that the characteristic peaks of the blank-loaded zein-SPI NPs are located at 3279 cm^−1^, 1639 cm^−1^, 1537 cm^−1^, corresponding to hydroxyl, amides I and II bonds. When Cur and Dio were loaded in the zein-SPI NPs, the hydroxyl peak shifted to 3280 cm^−1^ and 3282 cm^−1^, indicating that O-H stretching and vibration occurred when the drug was loaded into the NPs [34]. The peak of the amide I bond moved from 1639 cm^−1^ to 1634 cm^−1^, indicating C-O stretching with drug incorporation. The amide II bond shifted from 1537 cm^−1^ to 1538 cm^−1^, indicating C-N stretching and N-H bending [35,36]. Double peaks were observed in different groups of NPs, ranging from 2800 cm^−1^ to 2900 cm^−1^, which indicated CH_2_ stretching [37]. These results suggest that hydrogen bonding and electrostatic interactions were the main forces behind the formation of Cur- and/or Dio-loaded zein-SPI NPs. When Cur-loaded zein-SPI NPs were formed, the incorporation of Cur changed the protein configuration and the morphology of the amide II bond. This may be due to the presence of hydroxyl groups on Cur [38], allowing polyphenols to form hydrogen bonds with proteins, which is consistent with previous research results [11]. Proteins and polyphenols form covalent bonds (hydrogen bonds and hydrophobic interactions) [39]. When Dio and Cur were co-encapsulated into zein-SPI NPs, it could be seen that the intensity of their amide II bond peaks became lower, which also indicated that Cur was more hydrophobic than Dio, and the co-loading drug caused Cur to enter into a more hydrophobic microenvironment. 

### 3.4. Fluorescence of Free Bioactive Substance and Encapsulated Bioactive Substance

In the FTIR experiment, it was found that the co-encapsulation of Cur and Dio changed the structure of the formed protein’s NPs. When Dio and Cur were co-encapsulated, the strength of the NPs amide II bonds was significantly reduced, so it was judged that Cur was encapsulated in a more hydrophobic microenvironment. As shown in Figure 3B, the fluorophore was more sensitive to the surrounding environment, so this section analyzed the impact of the presence of Dio on the Cur microenvironment by fluorescence spectroscopy. The fluorescence spectra of Cur were selected to evaluate the encapsulation process of zein-SPI NPs [24]. First, it was observed that the fluorescence emission peak of a single Cur in 70% ethanol solution was 545 nm, and the position of the emission peak of Cur at the same excitation wavelength did not change after the addition of Dio. This also showed that the microenvironment of Cur was not altered with the simple addition of Dio. When Cur was encapsulated in zein-SPI NPs, Cur produced an emission peak of 535 nm, and with the addition of Dio, the emission peak of Cur shifted from 535 nm to 531 nm, and the fluorescence spectrum underwent a blue shift, indicating the change of the hydrophobic microenvironment in the formed NPs, which is also consistent with previous result obtained by FTIR studies. It was shown that Cur and Dio had different binding sites in the NPs, and Cur moved from a relatively polar microenvironment to a more hydrophobic core of zein NPs, which is consistent with the result obtained in previous studies. When resveratrol and Cur were co-encapsulated, Cur persisted in a more hydrophobic microenvironment [30]. At the same time, it can be observed that when the bioactive substance was encapsulated by NPs, its fluorescence intensity increased and the emission peak was blue-shifted, indicating an increase in the water solubility of the bioactive substance after encapsulation.

### 3.5. XRD

XRD was used to study the crystal structure and performance of free and bioactive substance-loaded NPs. It can be clearly observed from Figure 3C that both Cur and Dio had high crystallinity with the characteristic peaks of Cur at 12.1°, 14.7°, 17.0°, 18.3°, 21.3°, 23.5°, 24.8°, 26.4°, 28.4°, 29.2°, and the characteristic peaks of Dio at 12.6°, 14.2°, 18.6°, 21.2°, 15.8°, 27.1° and 28.2°, respectively. It is worth noting that the free bioactive species did not appear as any crystalline characteristic peaks in the bioactive substance-loaded zein-SPI NPs sample, indicating that Cur and Dio were distributed in an amorphous form in the formed zein-SPI NPs by hydrophobic interactions with zein and the soy protein isolate. However, as observed in Figure 3C, when Cur and/or Dio were incorporated with zein-SPI NPs, NPs showed a characteristically broad peak of around 20° [40]. These results indicated that free bioactive substances were successfully loaded into zein-SPI NPs, which also indicated that there were molecular interactions among zein, SPI, Cur, and Dio [41].

### 3.6. pH Stability

In order to extend the application of NPs in future commercial and various environments and to ensure the stability of NPs, the acid-base stability of Cur/Dio-loaded zein-SPI NPs was investigated. This provided some predictive help for the transportation and in vivo delivery of NPs loaded with bioactive substances in the future. In this section, bioactive-substance-loaded zein-SPI NPs were prepared using the pH-driven method. The isoelectric point (pI) of simple zein NPs was reported to be approximately 6.2 [42]. When the pH value was close to the pI, the zein NPs reduced the electrostatic force and the steric resistance due to the low-surface charge, resulting in the precipitation of NPs [26]. However, as shown in Figure 4A, the Z-average did not change much when the pH value was close to the isoelectric point of zein, illustrating the deposition of SPI on the surface of hydrophobic-zein NPs. The pI of SPI was around 4.5 according to Fan’s research [29]. When the pH value was changed to around 4–5, the bioactive substance-loaded zein-SPI NPs were found to exhibit unstable and significantly increased nanoparticle size. As shown in Figure 4B, at around pH 4 and pH 5, the zeta potential of NPs was close to zero. The weak electrostatic repulsion between the particles led to hydrophobic aggregation between the composite NPs. Obviously, the composite NPs were stable at pH < 4 and pH > 5, indicating that sufficient electrostatic repulsion can be generated between the NPs at the isoelectric point far from the outermost material of the composite NPs, in order to maintain their relative stability.

### 3.7. Bioactive Substance EE and LE

Compared with the traditional liquid–liquid dispersion method, Chen found that the EE value of nanocomposites constructed by the pH-driven method was 87.4%, which was higher than 85.7% by the liquid–liquid dispersion method, showing a better encapsulation effect [23]. In Feng’s experiment, under neutral pH 7 conditions, CS and zein were combined more tightly than that by the anti-solvent method. Under neutral conditions, the encapsulation efficiency of curcumin by the pH-driven method could reach 95%, no less than that by the anti-solvent method [43]. Therefore, to form nanoparticle complexes with smaller Z-averages and higher EE and LE percentages, the pH-driven method might be better than the other methods. Accordingly, the bioactive substance-loaded zein-SPI NPs were constructed by the pH-driven method to afford a tighter structure and higher EE and LE percentages. Moreover, the smaller nanoparticles facilitated cell absorption and increased the administration of bioactive substances [44]. It is found that the encapsulation of Dio in zein NPs by the pH-driven method is yet to be reported, and the related EE% and LE% changes after the co-encapsulation of Cur and Dio have never been explored previously.

In Table 1, 84.29% of Cur and 43.07% of Dio were encapsulated into the NPs carrier when zein-SPI NPs were used to encapsulate the bioactive substances. By co-encapsulation, the EE% of Cur decreased to 80.95%, which indicated that Cur and Dio had the same binding site, resulting in their competition for the same hydrogen bonds and hydrophobic binding sites on zein molecules. Interestingly, the EE% of Dio increased to 73.41% after co-encapsulation of Cur and Dio, which might be due to the π-π stacking phenomenon in the benzene ring structure of polyphenols [45]. Such a phenomenon was also found in Yu’s experiment [46], where the loading of icaritin and doxorubicin increased significantly when the molar ratio of icaritin to doxorubicin was 1:2. It is also possible that the introduction of Cur under alkaline conditions changed the configuration of the protein, which was conducive to the further binding between Dio and the protein. Previous studies have shown that, via the antisolvent method, the Cur’s EE% was lower than that by the pH-driven method, indicating a change in protein configuration under alkaline conditions [26]. For the co-encapsulated Cur and Dio, the LE% value increased to 16.54%, which indicated that the co-encapsulation of the two bioactive substances could help to increase the overall LE% value of NPs, resulting in higher bioavailability of bioactive substances; additionally, a slight increase in the Z-average of the NPs was found, further confirming the change in protein configuration. A similar phenomenon was produced in Hu’s study, in which the hydrodynamic diameter of quercetin and temozolomide increased somewhat after co-encapsulation [47]. In summary, the co-encapsulated Cur and Dio in zein-SPI NPs greatly promoted the EE% of Dio and increased the overall LE% value of NPs, while the increase of NPs Z-average and the slight change of Cur loss had little impact.

### 3.8. Storage Stability Also Related to Bioactive Substance

In this section of the experiment, the storage stability of Cur- and/or Dio-loaded zein-SPI NPs was considered. Due to the oxidative degradation of polyphenols, the aggregation and precipitation of NPs might occur, which can have a certain impact on the transport and commercial use of NPs. Therefore, the supernatant of the samples was tested four times within a month, with an average of once a week. It can be found from Figure 5A that the Cur-loaded zein-SPI NPs retained 88.09% of Cur after 4 weeks of storage, which was higher than Dio-loaded zein-SPI NPs, retaining 61.96% Dio. The main reason might be that Cur is more hydrophobic than Dio, which could be observed from the LogP value. The LogP value of Cur was 3.62, higher than that of Dio (3.06) (https://go.drugbank.com/) (accessed on 29 June 2023). Therefore, NPs could trap Cur into a more hydrophobic microenvironment, making Cur bind more tightly to zein-SPI NPs compared to Dio. It was found that in Cur/Dio-loaded zein-SPI NPs, Cur leaked faster than that in solely Cur-loaded zein-SPI NPs, which might be due to the competition between Cur and Dio for the same zein binding site [48,49]. However, an advantageous phenomenon was found in our experiment. As shown in Figure 5B, compared to Dio-loaded zein-SPI NPs, Cur/Dio-loaded zein-SPI NPs had a higher retention rate of Dio after 4 weeks of storage, increasing from 61.96% to 82.41%. This phenomenon was like previous studies, where the co-encapsulation of α-tocopherol and resveratrol increased resveratrol retention in storage [50]. In total, the NPs constructed by the pH-driven method could promote the hydrophobic structure of zein from the inside to the outside during acidification, rearranging the molecular structure of zein and promoting its binding to bioactive substances, thus increasing the LE% value of bioactive substances while inducing changes in the protein structure. Such changes might lead to an increase in the binding sites of bioactive substances, resulting in them binding more closely with zein, increasing the retention of bioactive substances and improving the storage effect.

### 3.9. Bioactive Substance Release

To make gastrointestinal digestion data more intuitive, the Z-average changes of different bioactive substance-loaded zein-SPI NPs at different digestion times were investigated to determine the physical stability of zein–NPs. Samples were taken every 30 min to measure the Z-average of the sample after digestion. As shown in Figure 6A, it was found that the Z-average in the stage of SGF had a certain increase, but it was still at the nm level (< 1 um). Considering the digestion by pepsin, the NPs underwent covalent bond disruption and hydrolysis of small molecule polypeptides [51]. However, with the change in the digestive environment, the Z-average increased rapidly after digestion sample transitioned from SGF into SIF. Considering that the pH changed drastically after SGF enters SIF, most of the polyphenolic substances such as Cur and Dio had high solubility in SIF, so a large number of bioactive substances were released and destroyed the structure of the NPs, resulting in a large increase in the Z-average. The presence of SIF pancreatic enzymes made the NPs quickly dissociate, reduced their surface charge, and caused the hydrophobic aggregation of NPs. With the extension of digestion time, the Z-average of NPs gradually decreased after reaching a certain peak. After the NPs were hydrolyzed by pancreatic enzymes, the generated peptides, proteins and bile salts repolymerized together with extending the digestion time to form micelle-like substances, so that the Z-average of NPs decreased again [52,53].

As most hydrophobic bioactive substances have higher solubility in SIF than in pepsinization, the release efficiency of different bioactive substance-loaded zein-SPI NPs at different digestion stages were measured to observe whether the bioavailability of bioactive substances was increased due to the formation of bioactive substance-loaded zein-SPI NPs. It is observed from Figure 6B that in experiments of simulating gastrointestinal digestion at the pepsin digestion stage, Dio-loaded zein-SPI NPs and Cur-loaded zein-SPI NPs only released about 22% of Dio and 18% of Cur. With the prolongation of digestion time, a large amount of release occurred when entering SIF, with the release rates of Dio and Cur increased to 57% and 35%, respectively, within 120 min. The release efficiency of Cur-loaded zein-SPI NPs was lower than that of Dio-loaded zein-SPI NPs, and the release rates of Cur- in Cur-loaded zein-SPI NPs were lower than that of Cur/Dio-loaded zein-SPI NPs. These phenomena could be explained by the following mechanisms: (1) Cur bounded to zein more strongly than Dio, resulting in its slow release, but it also showed that the bioavailability of Dio was higher than that of Cur. (2) The release of Cur from Cur/Dio-loaded zein-SPI NPs was higher than that of Cur-loaded zein-SPI NPs, mainly because Cur and Dio had the same binding sites. Cur produced a certain form of co-leakage during the digestion of SGF and SIF, resulting in excessive release of Cur, which was consistent with the previous experimental results shown in 3.8. There was a certain amount of Cur leakage when it was stored with Dio. It was found that the encapsulated of NPs had a certain slow-release effect in the gastrointestinal digestive environment compared with the pure bioactive substances, which controlled the premature release of the bioactive substance and improved its bioavailability, proving the impact of zein-SPI NPs encapsulation on the bioavailability of bioactive substances. In previous studies, the co-loading of Cur with different bioactive substances affected the physiological activity of Cur. The co-loading of Cur with EGCG increased the antioxidant activity and bioavailability of Cur [53]. The co-loading of Cur with piperine improved the storage stability and controlled-release efficacy of Cur [54]. The co-loading of Cur with apigenin enhanced their overall anti-inflammatory efficacy [12]. 

### 3.10. MTT

The successful application of a nanocarrier requires evaluating its biotoxicity. Accordingly, after the evaluation of its potential bioavailability, it is necessary to consider whether zein-SPI NPs themselves are biocompatible, so as to assess whether the nanocarrier entering the human body causes potential harm to human life safety. Therefore, samples at different concentrations (100, 200, 400, 800 ug/mL) were selected to evaluate the potential biocompatibility and determine the application value of NPs based on their toxicity to the human body. From the results shown in Figure 6C, the cell viabilities of all components were higher than 90% after the addition of MTT, indicating that zein-SPI NPs themselves are non-toxic and have good biocompatibility with the human body [2]. At the same time, it also showed that the NPs constructed by the pH-driven method had a certain safety in in vivo administration. Compared with the anti-solvent method, the advantage of the pH-driven method without organic solvents was one of the criteria for its great potential in the food industry.

### 3.11. Re-Dispersibility of zein-SPI NPs

When considering NPs with potential commercial use, cost is an important factor in terms of application and transportation. Therefore, how to redisperse powder after transportation became a gradual concern, especially when considering the transportation of freeze-dried powder. Due to the hydrophobic structure of the non-polar amino acids of the zein NPs, it was easy to produce leakages of bioactive substances and destroy the NPs themselves after freeze-drying. Therefore, in order to solve this problem, some stabilizers, such as polysaccharides and surfactants, are often used to stabilize the surface of zein NPs [55]. Some lyophilized protective agents, such as sucrose, are used to prevent the occurrence of aggregation, but this is still a huge challenge in the application and development of NPs [56]. In this section of the experiment, the re-dispersibility of bioactive substance-loaded zein-SPI NPs was evaluated by detecting their Z-average potential EE% after freeze-drying reconstitution. It can be observed from Table 1 that the redispersed NPs did not change much in terms of Z-average and zeta potential, indicating that the formed NPs had good re-dispersibility. The addition of SPI had a certain freeze-drying protective effect, which was similar to previous studies [57]. For NaCas, stable zein NPs can be redispersed in water. Although there was little change in EE% among the three NPs loaded with different bioactive substances, it is worth noting that the EE% of the bioactive substance-loaded alone decreased slightly more than that of the NPs co-encapsulated with bioactive substances, indicating that the retention of EE% after redispersion of the co-encapsulated NPs was better than that of the NPs loaded alone. This lyophilized redispersion treatment increased the application range of NPs, further promoting their medical, foods, and industrial applications.

## 4. Conclusions

Overall, zein-SPI NPs were prepared using the pH-driven method, and were used for the first time for the co-encapsulation of Cur and Dio. The micromorphological analysis showed that zein-SPI NPs are composed of hydrophobic zein and hydrophilic SPI shells. Through XRD experiment, it was found that Cur and Dio were amorphous in NPs, and the FTIR experiment showed that the structure of the NPs changed after the co-encapsulation of Cur and Dio. After the structural change of NPs, the co-encapsulated Cur and Dio improved the encapsulated efficiency and storage stability of Dio. The results of encapsulation efficiency and fluorescence revealed that Cur and Dio have different binding sites. Compared to Dio, Cur was loaded into a more hydrophobic microenvironment. The loading of the bioactive substances increased the sustained-release effect and improved the bioavailability. This study provides a certain theoretical basis for co-encapsulated bioactive substances. The superior loading and storage stability of bioactive substances are conducive to their future food applications and potential medical applications of NPs.
zein-SPI NPs were constructed by the pH-driven method with the optimum mass ratio (2:3) of zein to soy protein isolate to form a new plant–food–protein-based nanocarrier with the Z-average of 81.73 nm and Zeta-potential of −39.3 mV.The co-encapsulated Cur and Dio changed the structure of zein-SPI NPs, increased the binding sites of Dio in NPs, increased the EE% of Dio from 43.07% to 73.41%, and increased the overall LE% of NPs from 4.61% and 9.03% to 16.54%.The co-encapsulated Cur and Dio improved the retention rate of Dio in four weeks from 61.96% to 82.41%, making some progress in the field of co-encapsulated bioactive substances.In MTT experiment, HepG2 and L-O2 cells in different concentration groups (0–800 µg/mL) had more than 90% cell vitalities, indicating that zein-SPI NPs have strong biocompatibility and potential medical value. zein-SPI NPs were prepared using the pH-driven method and used for the first time for the co-encapsulation of Cur and Dio.

## Figures and Tables

**Figure 1 foods-12-02861-f001:**
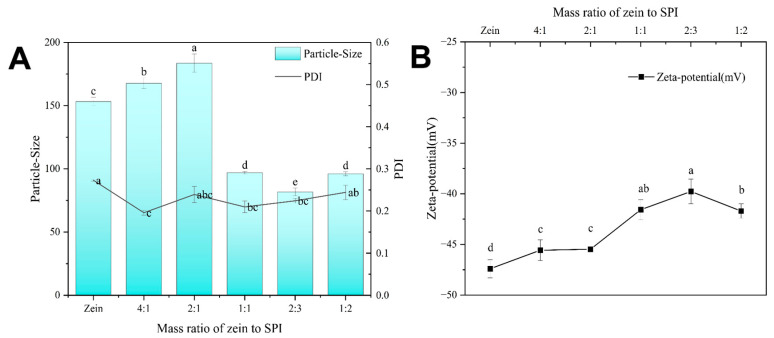
(**A**) Mass ratio of zein to SPI Z-average, (**B**) Zeta-potential of zein-SPI NPs. Data represent mean of three experiments. Different letters (a–e) mean data significance difference.

**Figure 2 foods-12-02861-f002:**
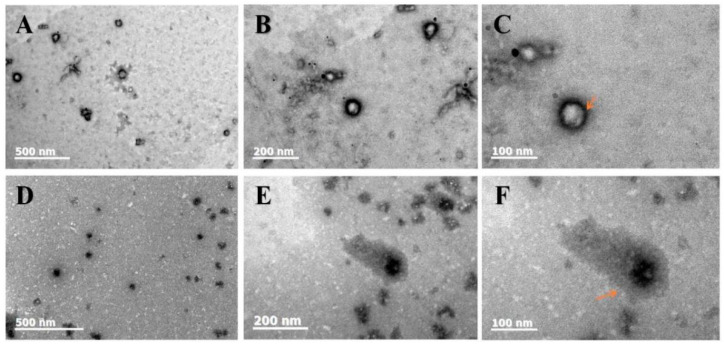
The TEM images of (**A**–**C**) Represent blank-loaded zein-SPI NPs and (**D**–**F**) Represent bioactive substance-loaded zein-SPI NPs under different scale bars from 500 nm to 100 nm. The arrow represent NPs shell shape.

**Figure 3 foods-12-02861-f003:**
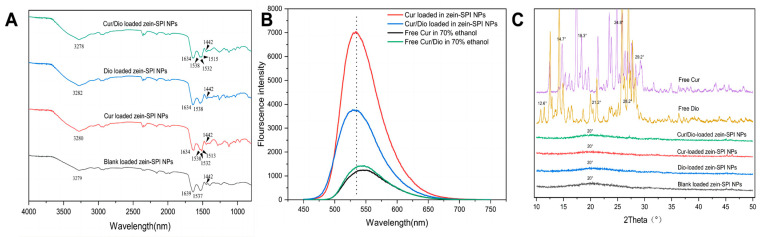
Different groups of zein-SPI NPs. (**A**) ATR-FTIR spectra of blank/bioactive-loaded zein-SPI NPs, (**B**) Free bioactive in ethanol and bioactive-loaded NPs, (**C**) XRD of free Cur, Dio and blank/bioactive-loaded zein-SPI NPs.

**Figure 4 foods-12-02861-f004:**
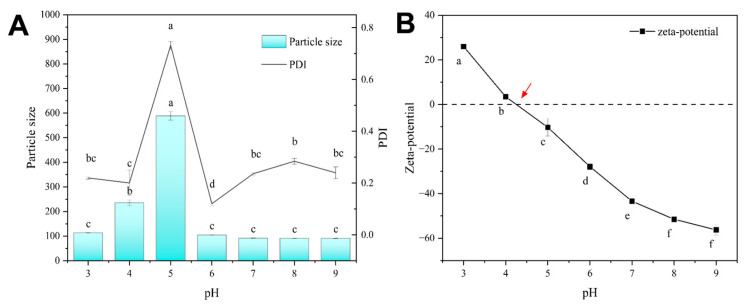
(**A**) The pH stability of bioactive-loaded zein-SPI NPs Z-average and PDI value, (**B**) the effect of bioactive-loaded zein-SPI NPs surface potential at different pH values. Data represent mean of three experiments. (The red arrow represents the Isoelectric point of zein-SPI NPs, different letters mean data significance difference).

**Figure 5 foods-12-02861-f005:**
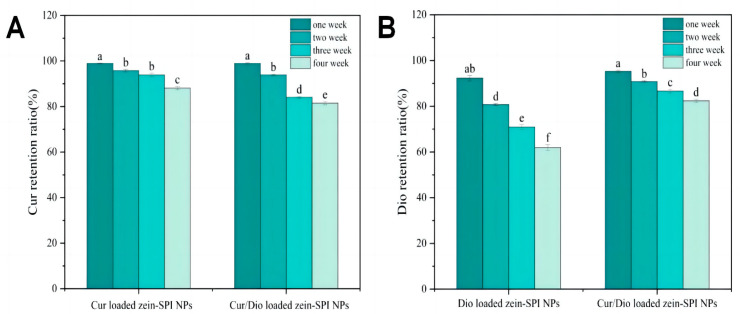
(**A**) The Cur retention ratios of Cur-loaded zein-SPI NPs and Cur/Dio-loaded zein-SPI NPs under four weeks storage, (**B**) The Dio retention ratios of Dio-loaded zein-SPI NPs compared with Cur/Dio-loaded zein-SPI NPs under four weeks storage. Data represent mean of four experiments. Different letters mean data significance difference.

**Figure 6 foods-12-02861-f006:**
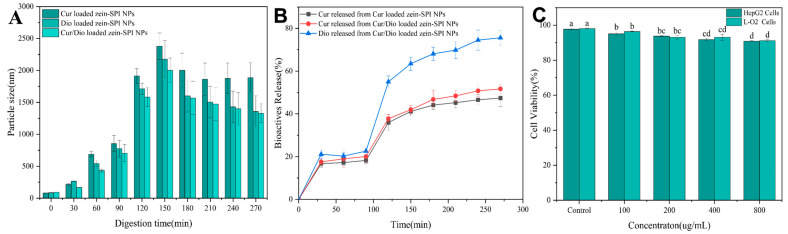
(**A**) The Z-average effect of different group of bioactive-substance-loaded zein-SPI NPs under gastro-intestinal digestion, (**B**) The bioactive substance released from bioactive substance-loaded zein-SPI NPs under simulated gastro-intestinal digestion condition (**C**) Different concentration of blank-loaded zein-SPI NPs to test HepG2 and L-O2 cell viability. Data represent mean of three experiments. Different letters mean data significance difference.

**Table 1 foods-12-02861-t001:** The bioactive substance-loaded zein-SPI NPs characteristics include Z-average, Zeta-potential, EE% and LE% before and after lyophilization. We need to pay more attention to the Z-average parameter and peak maximum in the droplet size distribution, versus the scattered light intensity mode, as this was not fully sufficient, and also look more closely at the results from mathematical calculations made by the software. But it can still be effective, as the real result error showed less than or equal to 1%.

Sample	Group	NPs Size (nm)	PDI	Zeta-Potential (mV)	EE (%)	LE (%)
Cur-loaded	Freshly	81.89 ± 0.38c	0.22 ± 0.01ab	−39.37 ± 0.22ab	84.29 ± 0.51a	9.03 ± 0.05b
	Re-dispersed	84.60 ± 0.78b	0.26 ± 0.01a	−38.37 ± 0.35a	78.91 ± 0.76b	
Dio-loaded	Freshly	86.21 ± 0.98b	0.20 ± 0.02b	−39.07 ± 0.48ab	43.07 ± 1.55d	4.61 ± 0.17c
	Re-dispersed	86.24 ± 0.70b	0.22 ± 0.02ab	−39.60 ± 0.23ab	37.46 ± 1.38e	
Cur/Dio-loaded	Freshly	91.45 ± 0.64a	0.19 ± 0.01b	−40.10 ± 0.55b	80.95 ± 1.36ab (Cur)/73.41 ± 1.72c (Dio)	16.54 ± 0.61a
	Re-dispersed	93.23 ± 0.43a	0.20 ± 0.01b	−39.37 ± 0.38ab	80.48 ± 1.22ab (Cur)/72.40 ± 0.95c (Dio)	

Different letters representative significance difference no need change.

## Data Availability

The datasets generated for this study are available on request to the corresponding author.

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
