# Peer review of "Co-Encapsulation of Curcumin and Diosmetin in Nanoparticles Formed by Plant-Food-Protein Interaction Using a pH-Driven Method"

_foods, 2023, doi:10.3390/foods12152861_

Round 1

Reviewer 1 Report (New Reviewer)

The Authors presented the results of capsule development for curcumin and diosmetin. The results are interesting and suitable for Foods. Minor revision is needed. Detailed comments to the Authors are listed below:

Abstract

- what do you mean that zeta stabilized?

- what do you mean by co-delivery?

- the size of the composition, especially optimal composition is mandatory in the abstract.

Introduction

- for the encapsulation of various compounds some interesting materials you can find here DOI:10.1016/j.eti.2023.103278; doi.org/10.1515/rams-2022-0282; doi/10.1515/ntrev-2014-0018/html Especially the last one is interesting due to curcumin loading

- the sentence “Subsequently, the 141 potential of composite…” is difficult to catch the meaning. Due to the large introduction, you can remove it or improve the meaning

Materials and methods

-        Subsection 2.2 It will be easier to name it “samples preparation”

-        Line 175 what do you mean by “aliquot”

-        What were the scattering angle and temperature in DLS measurements?

-        Just a comment – three repetitions for the sample analysis is not enough. For further examination, it is recommended to replicate DLS 5-10 times, especially for the samples with high PDI.

-        Did you dry the samples for TEM or use cryo mode? Please add some more details.

-        Detailed samples description presented i.e. in the table is mandatory.

Results and discussion

-        Please specify what parameter that you name particle size did you determine by DLS? Z-ave, mean peak size (by intensity or by number) – please have a look at how it should be described i.e. here https://doi.org/10.1016/j.fbio.2023.102559

-        The plot for Zeta is incorrect – please left is the points or bars

-        Please enlarge Figure 3 before the final presentation

-        If possible please separate the XRD spectra and add some comments on the figure where typical peaks can be identified

-        The comparison of pH versus PDI and zeta is interesting

-        Table 1 – please have a look for DLS accuracy (some discussion you can find in recommended papers) typically it is 1nm and 0.1 for proteins the sizes 0. xx it is below the DLS accuracy (the software just gives you statistic) consider to improve the results presented in Table 1 or add some additional comments in the body text.

-        Figure 6 please consider uniforming the colors in the graphs.

-        Please add some images and photos of as prepared systems.

  •  

Author Response

Reviewer 2 Report (Previous Reviewer 2)

This research was carried out on "Co-encapsulation of curcumin and diosmetin in nanoparticles formed by plant-food-protein interaction using pH-driven method". Although it is a very successful research, it needs minor revision on the following points.

1.    Please add your method to the keywords

2.    When ethanol solvent is used in certain proportions, it has the advantages of being used in edible film synthesis.

3.    Please specify other chemicals in detail in the material section and add their chemical properties.

4.    2.2. Please add how many degrees the drying process was carried out in the lyophilizer in the title

5.    2.7.2. In MTT assays, cell viability was incubated for only 4 h, whereas a period of at least 24 h is generally used in the literature. Please refer to the literature about the duration by stating the reason.

6.    Please refer to the literature by making detailed comments for the 2800-2900 cm-1 spectra in the FTIR analysis results. The following article will help you.

Encapsulation and antibacterial studies of goji berry and garlic extract in the biodegradable chitosan

G Baysal, HS Olcay, Ç Günneç

Journal of Bioactive and Compatible Polymers 38 (3), 209-219

Author Response

请参阅附件

This manuscript is a resubmission of an earlier submission. The following is a list of the peer review reports and author responses from that submission.

Round 1

Reviewer 1 Report

Dear authors, 

Please consider the following suggestions to improve the manuscript:

L22-23 - Turmeric (Curcuma longa L.) is traditionally used in ayurvedic medicine.

L25 - ''Dio'' - what is the meaning since the authors did not explained the acronym in the text. Rewrite as : Diomestin (Dio). The same issues in lines 50 (for NP) and 59 (SPI).

L34 - I am not sure if ''fight'' is a correct term. Explain how curcumin coencapsulated with quercetin showed anti-cancer potential.

L51-53 - Where/who such research was done? It is not clear the way it was written.

L55-57 - Any idea why? Include in text possible causes why fucoidan and zein overcame stability of NP.

L62 - BSA was not defined.

L69-L73 The novelty of manuscript is still unclear. I recommend authors rewrite introduction mentioning as well the advantages of pH-driven method in comparison to other methods of encapsulation and why curcumin and diomestin were selected. Include also recent works that support the hypothesis stated by authors.

L89 - It is Citric, not ''critic''

L90 - ''more concrete''? What do you mean?

L94 - To which pH were the dispersions adjusted? From 12 to 7? 

L96 - Include the brand, model, city and country of freeze dryer used.

L119 - What was the dilution rate used?

L133 - Which article? Include references

L141-143 - Insert method specifications (flow rate, injection volume, wavelenght; concentrations of curcumin and dismestin used for stdandard curve...)

L144 - What is EE? Acronym was not explained.

L145 - Include equation used to calculate encapsulation and loading efficiencies.

L156 - Which previous procedures? Where are the references?

L274 - Which figure?

Table 1 - Insert the statistical differences. Looks like there are no differences in particle size, PDI and zeta potential within fresh and redispersed samples.

L355-L359 - Sounds contradictory mentioning logP differences in line 355 and then mention they are similar in line 359.

L363-364 - How does it make sense? The aggregation is just a physical destabilization of a system. How the antioxidant potential of curcumin may be related to that since it was not measured in here?

Figures 1;4;5-6: Express the statistical difference.

Additional comments:

1 - in M&M section I recommend the author include a topic explaining how the pH-driven method experiments were done. It sounds like the 2.2 topic is a different thing.

2- The authors need to provide a precise discussion by comparing your results with literature. For instance, in lines 312-313 the discussion is superficial and the EE/LE percentages of anti-solvents and liquid-liquid dispersion methods were not provided, but mere citations without explaining what is going on.

3 - ''In vitro'' is a latin expression, so must be written in italic.

4 - Resolution of images are not good. Also, for graphics I recommend use Arial or calibri instead of Times new roman.

L100 - The sentence does not make sense. Since the samples are supposed to be analyzed it is obvious that a blank must be included as control.

L-101-102 - Instead of ''Before..., take 1 mL'' use: Before the analysis, 1 mL of sample aliquot was taken and mixed with ultrapure water at a 1:10 (v/v) ratio

L108-109  - The ''after dilute prepared finished was'' does not make any sense. Instead of  ''The sample... was'' try use: The diluted nanodispersions were prepared..

Reviewer 2 Report

This manuscript is research on “Co-encapsulation of curcumin and diosmetin in nanoparticles formed by plant protein interaction using pH-driven method”.  the research is  quite impressive, but unacceptable without minor revision. comments are attached as a file

1.    Please share your analysis findings in the abstract section, especially your MTT findings in the conclusion section, and your encapsulation efficiency

2.    I recommend that you add a comparison table containing the analysis results of the literature findings of the last five years to the discussion section.

3.    Curcumin is an outstanding bioactive ingredient and many studies have proven how effective it is in medical applications. The following article will help you

Biocidal Activity of Bone Cements Containing Curcumin and Pegylated Quaternary Polyethylenimine

T Eren, G Baysal, F Doğan

Journal of Polymers and the Environment 28, 2469-2480

4.    please improve similarity ratio in characterization sections

5.    In the conclusion, please highlight the innovative propositions for this research more clearly

6.    Minor editing of English language required

Minor editing of English language required

Round 2

Reviewer 1 Report

Dear authors,

The authors addressed the reviewer's comments and improved considerably the quality of manuscript. Please consider the following:

L129-130 - The writing is difficult here  Instead of ''In Chen article...'', the authors should rewrite as: Chen and coworkers [26] observed that co-encapsulation of resveratrol with Cur, resulted in extended  stability of resveratrol... - Make sure the writing makes sense for this and other sentences throughout the text.

L131 - envisaged? You mean, hypothesized?

L136-142 - Please rewrite the sentence starting as: For this reason, this work investigated the co-encapsulation of Cur and Dio with the use of zein and SPI as wall material. The pH-driven method was used to prepare the particles....

I recommend the authors improve the quality of Figure 3. Instead of writing in times new roman, please use arial or calibri fonts, so there will be no issues in writing resolution.
